# HIV/STI/HCV Risk Clusters and Hierarchies Experienced by Women Recently Released from Incarceration

**DOI:** 10.3390/healthcare11081066

**Published:** 2023-04-07

**Authors:** Karen A. Johnson, Timothy Hunt, Lisa Puglisi, Ben Chapman, Amali Epa-Llop, Johanna Elumn, Peter Braick, Navya Bhagat, Elizabeth Ko, Antoinette Nguyen, Rachel Johnson, Heather K. Graham, Louisa Gilbert, Nabila El-Bassel, Diane S. Morse

**Affiliations:** 1School of Social Work, University of Alabama, Tuscaloosa, AL 35487, USA; 2Social Intervention Group, School of Social Work, Columbia University, New York, NY 10027, USA; 3SEICHE Center for Health and Justice, School of Medicine, Yale University, New Haven, CT 06520, USA; 4Department of Psychiatry, University of Rochester School of Medicine, Rochester, NY 14642, USA; 5Educational Studies in Psychology, University of Alabama, Tuscaloosa, AL 35487, USA; 6Department of Medicine, University of Rochester School of Medicine, Rochester, NY 14642, USA

**Keywords:** HIV, hepatitis C, STIs, SAVA syndemic risks, homelessness, depression, PTSD, recently released women

## Abstract

This study examines cross-sectional clusters and longitudinal predictions using an expanded SAVA syndemic conceptual framework—SAVA MH + H (substance use, intimate partner violence, mental health, and homelessness leading to HIV/STI/HCV risks)—among women recently released from incarceration (WRRI) (n = 206) participating in the WORTH Transitions (WT) intervention. WT combines two evidence-based interventions: the Women on the Road to Health HIV intervention, and Transitions Clinic. Cluster analytic and logistic regression methods were utilized. For the cluster analyses, baseline SAVA MH + H variables were categorized into presence/absence. For logistic regression, baseline SAVA MH + H variables were examined on a composite HIV/STI/HCV outcome collected at 6-month follow-up, controlling for lifetime trauma and sociodemographic characteristics. Three SAVA MH + H clusters were identified, the first of which had women with the highest overall levels of SAVA MH + H variables, 47% of whom were unhoused. Hard drug use (HDU) was the only significant predictor of HIV/STI/HCV risks in the regression analyses. HDUs had 4.32-fold higher odds of HIV/STI/HCV outcomes than non-HDUs (*p* = 0.002). Interventions such as WORTH Transitions must differently target identified SAVA MH + H syndemic risk clusters and HDU to prevent HIV/HCV/STI outcomes among WRRI.

## 1. Introduction

Yearly, 1.9 million American women are released from incarceration in the United States [1], a significant percentage of whom have a high need for intervention, treatment, and support for substance use and intimate partner violence (IPV). The prevalence of substance use disorder (SUD) and IPV in this population has been noted to be as high as 51% [2] and 90%, respectively [3,4,5,6], as compared to 7.7% [7] and 3% in the general population, respectively [8]. In addition, approximately 30% of incarcerated women are believed to have an alcohol use disorder, as compared with a rate of 24% among all incarcerated individuals [2]. A cyclical association has also been identified between IPV, trauma, and the misuse of substances (e.g., alcohol, cocaine, cannabis, and opioids) [6], and between the severity of trauma and SUD among recently released women in particular [9]. Substance use and IPV are also deeply intertwined with poor decision-making, increased risk-taking, and adverse health consequences among justice-involved women, all of which are linked to HIV transmission, other sexually transmitted infections (STIs) [10], and Hepatitis C (HCV) [11].

Significantly compounding these challenges, rates of mental health disorders and homelessness are also disproportionately high among women emerging from jails and prisons. The co-occurrence of SUD and mental health disorders among women with histories of incarceration ranges from 26% for the co-occurrence of alcohol use disorder and a mental health diagnosis, to 10% for the co-occurrence of methamphetamine use disorder and a mental health diagnosis among those with an opioid use disorder (OUD) [7]. In addition, more than 20% of women with histories of incarceration have been found to have one or more co-occurring severe mental illnesses and substance use disorders (SUDs) [12], compared with 5.9% among males [13]. In terms of homelessness in particular, approximately 2.7% of women with histories of incarceration have been identified as homeless [14], as compared with 0.13% among women without similar backgrounds [15,16]. Women emerging from jails and prisons are more likely to experience homelessness than formerly incarcerated men, at a rate of 264 per 10,000 vs. 195 per 10,000 [14]. Unhoused women with histories of incarceration also have significantly higher histories of physical, sexual, and injurious assault from intimate and non-intimate sexual partners across the course of their life, along with co-occurring substance use disorders [17]. Without secure housing, recently released women are also at increased risk of relapsing into addiction [18] and IPV [19].

Among women with histories of incarceration as a whole, substance use disorders, intimate partner violence, mental health disorders (MHDs), and homelessness are oftentimes comorbid and exacerbate one another, and they can occur before, during, and/or after periods of incarceration [20]. They also lead to disproportionately high HIV and other sexually transmitted infection (STI) outcomes [9], and they are significantly associated with the transmission of HCV [21], with as many as 14% of women transitioning from carceral settings having been identified as having HIV [4,5,22], as opposed to the 0.7% rate found among the general population of women in the U.S. [23]. The need to close this gap remains urgent. Rates of hepatitis C virus (HCV) among women in jails and prisons have also been noted to be 1.21 times higher than the rates identified among men [24], further underscoring the need for targeted interventions and additional scholarly research.

Despite this gap, not enough is known about how SUDs, IPV, mental health disorders, and homelessness may synergistically combine to lead to disproportionately high rates of HIV, STIs, and HCV post-release. This is despite a vast body of peer-reviewed literature pointing to a significant synergistic impact of SUD and IPV on impoverished women, leading to an excess burden of HIV [5,9,25]. Referred to as the SAVA syndemic, scholars have suggested that this conceptual framework may serve as an important lens through which to understand risk synergism among women in the criminal legal system [26] and may intersect with mental health disorders [27] and homelessness [28]. To date, however, no studies have examined this intersection among women recently released from incarceration, representing a significant gap in the literature. Although whether and how mental health disorders and homelessness may combine synergistically with SAVA to increase HIV, other STIs, and HCV among recently released women has been neglected in the peer-reviewed literature, higher SAVA syndemic risks have been identified among recently released women with high rates of arrests (defined as four arrests or more), histories of childhood sexual abuse trauma, or who expressed a desire (or recognized a need) to change their sexual and drug-using behaviors [29]. Scholars have also underscored a need to understand how the SAVA syndemic and other behavioral health risks may differently cluster to create risk typologies [30].

Notwithstanding these gaps, a handful of cluster analytical approaches conducted to date offer some preliminary insights. First, hierarchical cluster analysis has demonstrated the need for trauma-informed services among formerly incarcerated people living with HIV [31]. Second, cluster analytic approaches used in combination with binary logistic regression identified a sub-cluster of individuals in the criminal legal system deemed to be frequent systems utilizers at high risk of reincarceration, with a shared risk profile of higher numbers of homeless episodes, rates of bipolar disorder and depression, and felony convictions [32]. While these studies have aided in efforts to identify intragroup risks among populations involved in the criminal legal system, no studies to date have utilized these methodological approaches to determine (1) what, if any, subgroups of women transitioning out of carceral settings may exist by SAVA, mental health risks, and homelessness experienced; and (2) which of these risks, if any, may be the most predictive of HIV/STI/HCV outcomes.

This study serves as an initial step towards addressing these gaps. Utilizing an expanded SAVA syndemic conceptual framework that includes mental health (severe depression, PTSD) and homelessness (MH + H), we sought to identify distinct clusters of common characteristics among recently incarcerated women according to the SAVA MH + H syndemic risks experienced. We also sought to examine which of these risks significantly predicted HIV/STI/HCV outcomes in this highly vulnerable population during the high-risk window after release. In identifying SAVA MH + H syndemic cluster profiles, we leveraged cross-sectional data collected from cisgender women (n = 206) participating in the WORTH Transitions program, all of whom were released from incarceration within 12 months of enrollment in the study. To determine longitudinal relationships between the SAVA MH + H syndemic risks experienced at baseline and future HIV/STI/HCV risks, we utilized data collected six months post-enrollment. Our findings build on the existing literature and deepen the understanding of differing HIV/STI/HCV risk profiles. The results also address a significant public health need to develop tailored HIV/STI/HCV prevention and intervention approaches guided by group differences and the main drivers of HIV/STI/HCV outcomes. Increasing our understanding of these heretofore understudied SAVA MH + H syndemic risk clusters and outcomes is a critical next step in ending the HIV epidemic and co-occurring sexually transmitted infections and HCV in this high-need, high-risk population during the especially vulnerable period post-release.

## 2. Methods

### 2.1. Intervention

WORTH Transitions combines two evidence-based interventions: Women on the Road to Health (WORTH) [22], and the Transitions Clinic Network Model [33,34,35]. WORTH is a group-level, five-session prevention intervention that has proven to be effective in reducing HIV, other sexually transmitted infections, HIV risk behaviors, and IPV among women in community corrections (e.g., probation, parole) who misuse substances. The Transitions Clinic Network Model provides culturally specific primary care strengthened by community health workers with lived experience of incarceration to individuals released from carceral settings, who are known to have almost 13 times greater morbidity and mortality rates than the general population. WORTH Transitions aimed to investigate whether a combination of both of these evidence-based approaches would improve the medical, substance use, mental health, and HIV/HCV/STI risk outcomes of the participants.

### 2.2. Recruitment and Enrollment

The recruitment and enrollment procedures are described in detail in an earlier manuscript [36]. Participants were recruited by formerly incarcerated peer community health workers using face-to-face and indirect (e.g., social media, flyers) recruitment methods from jails and prisons, as well as other areas where recently released women would visit and/or receive services. A total of 208 participants were enrolled across both sites; 206 completed baseline measures, and 101 completed 6-month follow-up surveys. The inclusion criteria were as follows: participants had to be female (transgender females permitted), released from incarceration within the past year, at least 18 years old, English-speaking, and have a (self-reported) recent history of alcohol and/or substance use disorder. The participants were asked to complete an initial screening, a one-on-one WORTH HIV/STI/HCV prevention session (which included HIV/HCV rapid tests and pre/post-test advice), four additional WORTH sessions, and one or more appointments with a primary care provider (PCP)—whether their own or one provided by either of the WORTH Transition clinics. The PCP appointments allowed for subsequent HIV/STI/HCV testing and treatment as required. WORTH individual and group sessions were delivered by trained, formerly incarcerated community health workers (CHWs). Appointments at the Transition Clinic were facilitated by a lone study physician (PI or co-I). Self-reported sociodemographic data along with self-reported and biologically confirmed data on HIV/STI/HCV risks and risk behaviors were collected at baseline and at 6-month follow-up.

### 2.3. Measures

All explanatory variables (i.e., substance use, IPV, lifetime experience of sexual and physical abuse) used in the clusters and regression were collected at baseline. Outcome variables (HIV/STI/HCV and related risk behaviors) were gathered in the 6-month follow-up survey.

#### 2.3.1. SAVA Syndemic Explanatory Variables

Substance use: This domain was represented by a count of how many substances were used in the previous 90 days out of 13 substances (i.e., crystal meth, hallucinogens, PCP, methadone, heroin, cocaine, crack, marijuana, anti-anxiety drugs, downers, and painkillers). The categorizations were as follows: 0, 1–3, or 4 or more (for a maximum of 8).

Intimate partner violence: Intimate partner violence was modeled using the Ongoing Abuse Scale (OAS)—a five-item, yes/no, self-report scale that measures physical, sexual, and emotional intimate partner violence at the present time. The OAS has previously been tested in various settings with different populations, including African Americans, Latino/as, and Whites (Cronbach’s alpha of 0.59) (54).

#### 2.3.2. Lifetime Trauma

Traumatic events: Utilizing the Lifetime Stressors Checklist-R (LSC-R) [37], trauma was modeled in two ways. The LSC-R assesses traumatic events over the course of the lifetime as a 30-item self-report measure. Items on the checklist include traumatic experiences that involve medical, financial, relationship, and family trauma, as well as traumatic experiences with physical and/or sexual abuse and/or assault. A summary variable was calculated based on the sum of positive answers in the LSC-R, in accordance with previous research [38]. This score represents the breadth of trauma exposure (that is, how many types of trauma a person has experienced); it is not intended to assign different weights to different traumas. For this reason, this score may not reflect trauma accumulation. In addition, it should be noted that the item pertaining to experiences with natural disasters was eliminated for the purposes of this study. The LSC-R was first reviewed as a composite variable ranging from 0 to 29, where the latter score would indicate that the participant had experienced all of the events. Second, we recoded some of the items in the LSC-R by clustering them into groups of trauma types. The first five clusters were coded as continuous variables: physical abuse/assault, sexual abuse/assault, family, medical, and the system. The remaining four clusters were coded as binary, single-item variables (i.e., the items emotional abuse, financial, relationship, and other events that the participant might choose to fill in). The LSC-R has been utilized with the criminal legal system in the past and demonstrated moderate-to-good test–retest reliability, as well as good convergent and criterion validity with PTSD symptoms in women infected with HIV [39].

#### 2.3.3. Covariates

Alcohol use: Alcohol use was categorized as the number of days (out of the previous 90) during which the participant used alcohol. This was categorized as follows: 0 days, 1–30 days, and >30 days.

Depressive symptomatology: Depressive symptomatology was categorized as an ordinal/categorical variable in accordance with the Global Assessment of Individual Needs (GAIN) scale. The GAIN scale is a 113-page validated assessment for the treatment, planning, and monitoring of persons with SUD, yielding a composite score of need, access, and utilization. Divisible into shorter sub-scales, the GAIN Internal Mental Distress Scale (IMDS) is reliable and well validated, thus, was used to assess depression and anxiety symptoms (categorized as severe, none, or moderate) over the previous 90 days and the previous year [40]. Post-traumatic stress disorder symptoms: PTSD was categorized as either negative or positive (i.e., a dichotomous variable) and used the Trauma Symptom Scale component of the IMDS. The IMDS reliably measures PTSD (α = 0.96).

Sociodemographic characteristics: Several sociodemographic variables were also included in the logistic regression model. These variables were as follows: (a) relationship status (categorized as single, in a relationship, or separated/divorced/widowed); (b) age (categorized as under 30, 30–40, or over 40), which was chosen based on the data distribution; (c) educational level (categorized as less than high school (HS), HS/General Educational Development (GED), some college, or college+); (d) race/ethnicity (categorized as White, or Black and Indigenous Women of Color (BIWOC)); (e) homelessness status (categorized as homeless or not homeless); (f) employment status (categorized as unemployed or underemployed, employed full- or part-time, or unknown/not reported); and (g) ever in foster care or adopted (categorized as no or yes).

#### 2.3.4. Outcome Variable

HIV/STI/HCV risks: HIV/STI/HCV risk outcomes were measured at the 6-month mark and modeled as a composite variable. This variable was assessed using self-reported risk behaviors such as unprotected sex, risky needle use, sex with someone who injects drugs, sex while high, sex with a male sexual partner who was HIV+, etc. Participants who endorsed any risk behavior, regardless of self-reported HIV/STI/HCV status, received a maximum of one point. Persons with self-reported HIV/STI/HCV status also received a maximum of one point, regardless of the amount of self-reported risk behavior.

−Risk behaviors were measured using the GAIN HIV risk scale (α = 0.96).−Self-reported HIV/STI/HCV was recorded as “yes” upon receipt of a positive answer to the question “In the past 6 months, have you been diagnosed with a sexually transmitted disease/HIV/HEP C?” [40].

### 2.4. Statistical Analyses

Statistical analyses were performed using the STATA statistics software package, version 15.1.

### 2.5. Bivariate Analyses

The composite HIV/STI/HCV risk outcome variable was examined in bivariate analyses with the expanded SAVA syndemic risk variables (hard drug use past 90 days, prescription drug use past 90 days, physical abuse aged >16, sex coerced aged >16, sex forced aged >16, mental health (severe depression, PTSD), and homelessness), and covariates, to identify significant associations.

### 2.6. Clusters

After basic descriptive statistics, we examined the variable distributions in preparation for clustering. The SAVA variables were characterized by large numbers of zero responses (i.e., the factor was absent), coupled with a smaller number of often extreme positive responses. To prevent undue influence of outliers on the cluster solution, these variables were categorized into presence/absence. Homelessness was already binary, and race was already characterized by two dominant categories (Black and White) and, therefore, also coded as binary. To place age and education on the same binary scale, we selected the split producing the greatest variation (>40 vs. less and high school or less vs. some college or greater, respectively); as variability increased, the likelihood variables differentiate between clusters [41]. Regarding violence variables, only 23% of the sample reported being in a romantic relationship, so items asking specifically about romantic partner violence had correspondingly low prevalence and low variability. Therefore, we represented the “violence” aspect of SAVA with the items about general physical and sexual abuse in childhood and adulthood. On the DSS, only 11% of the sample reported no depression symptoms (score of 0), while 14% reported mild symptoms (scores of 1–2), and 75% of the sample reported high levels of depression (scores of 3+). To capture the greatest binary variation for clustering, we transformed this to “high depression” vs. mild or none. PTSD was coded as a positive screen vs. negative screen.

We then computed a distance matrix based on the binary matching coefficient [42] and proceeded with k-means clustering, examining solutions from 2–6 clusters. We evaluated clusters using the mean Calinski–Harabasz pseudo-F statistic across 100 random starting values, higher values of which indicate more distinct clustering [42]. We also evaluated the theoretical and clinical utility of differing cluster solutions. Solutions with too few clusters might represent an overly simplistic typology that might not accommodate the variety of different SAVA profiles, while too many clusters could create an unwieldy typology.

After determining the optimal cluster solution, we identified individuals on the boundary between clusters based on low silhouette distance [43]. Silhouette distance ranged between −1 and 1, with −1 indicating that an individual is extremely similar to the nearest neighbor cluster and very dissimilar to the cluster to which they have been assigned (essentially, a poor classification). Values of 1 indicate that the individual is maximally similar to others within their cluster and maximally dissimilar to the nearest neighbor cluster. Values of zero indicate cases that are exactly on the boundary between two clusters. We pruned the final clusters by removing individuals with silhouette distances below 1, which eliminated cases close to the boundaries without removing too many people from the final cluster classification.

After deriving the final cluster classification, we computed the prevalence of each characteristic by cluster. Statistical tests of the differences are sometimes considered to be uninformative because the clustering algorithm, by design, identifies groups that differ in these factors; thus, it would not be surprising to see significant differences [42]. However, the magnitude of differences is descriptively informative and helpful in characterizing the groups.

### 2.7. Logistic Regression

Finally, we fitted a multivariate logistic regression, with an outcome of any HIV risk behavior, HIV diagnosis, or other STI diagnosis (vs. none) regressed on the predictors discussed above.

## 3. Results

Bivariate analyses: Alcohol use in the past 90 days, hard drug use in the past 90 days, coerced sex aged <16, forced sex aged <16, physical abuse aged >16, and forced sex aged >16 were significantly associated with the HIV/STI/HCV composite risk variable as an outcome.

Table 1 shows the Calinski–Harabasz pseudo-F across 100 random starts for each number of clusters. Table 2 shows the baseline descriptive statistics for the SAVA syndemic and sociodemographic variables. Fewer clusters generally resulted in higher pseudo-F values. Based on these results, we examined the two- and three-cluster solutions for further consideration. The two-cluster solution of one group consisting of about one-third of the sample showed markedly higher prevalence on all SAVA variables and lower rates of “at least some college” education, compared to the other two-thirds of the sample. The three-cluster solution was somewhat more differentiated (described below), and the average silhouette distances were comparable across both solutions (0.2). Thus, we proceeded with the three-cluster solution.

Twenty-five boundary cases were identified in this cluster solution with silhouette distances of 0.05 or less, roughly evenly across the three clusters, and removed to improve cluster homogeneity and differentiation; this raised the pseudo-F to 20. Table 2 shows the prevalence of characteristics across the three clusters after the removal of the boundary cases.

It should be noted that statistical comparisons across the clusters are meaningless, since clustering algorithms—by design—find groups that differ in as many of the variables as possible. However, it is possible to characterize the clusters descriptively based on their standing with respect to the different variables. Cluster 1 had higher levels of depression, PTSD, drug use, HIV risk behaviors, and childhood abuse than the other two clusters. Roughly 47% of the women in Cluster 1 were homeless. This was also the smallest cluster. Cluster 2 was slightly larger and showed relatively less drug use and higher numbers of women with at least some college education, with 40% of the women being homeless. Cluster 3 was the largest, with 32% of its members homeless, only 19% with college or greater education, and relatively fewer women with histories of abuse than the other two clusters.

### Multivariate Logistic Regression (Longitudinal)

All non-HIV variables in Table 2 were then used to predict a composite outcome of any HIV risk behavior, STI, HIV, or HCV. This analysis used all 153 individuals who fit cleanly into a cluster, as well as the 25 boundary cases that were equidistant between clusters, for a total N of 178; 42 were positive for the outcome. All variance inflation factors were <2 and tolerances were >0.59, indicating no multicollinearity [44]. Table 3 shows the logistic regression results that included all SAVA syndemic risk variables and covariates on HIV/STI/HCV outcomes. Hard drug use was the only significant predictor of the HIV/STI/HCV composite outcome, with hard drug users showing over four times the odds of HIV/STI/HCV risk compared to non-users. Coerced sex during childhood also showed a trend-level association with the outcomes, with those having experienced coerced sex having over twice the odds for HIV/STI/HCV risk behaviors.

## 4. Discussion

This is the first study to examine HIV/STI/HCV risk outcomes among women recently released from jails and prisons through an expanded SAVA syndemic lens that includes mental health risks and homelessness (SAVA MH + H). Several important findings were identified that have implications for tailoring SAVA syndemic, mental health, and homelessness interventions and services to this population of highly vulnerable women in order to prevent new transmission among them. First, across all three clusters identified, substantial differences were identified in the levels and types of SAVA syndemic, mental health, and homelessness risks. Despite being the smallest of the three, the first cluster (Cluster 1) was characterized by the highest levels of SAVA MH + H syndemic risks when compared with the other two clusters. Roughly 50% of women in this cluster were unhoused, and close to 100% self-reported using hard drugs, experiencing severe depression, and PTSD symptomology. Women in this cluster also reported experiencing the highest levels of childhood trauma (physical abuse, coerced sex, and forced sex) and physical abuse after the age of 16. This group was also characterized by the highest levels of HIV/STI/HCV risk outcomes (e.g., unprotected sex in the past 30 days and sex while high, with an HIV+ partner, or with an IDU partner in the past 30 days). Despite these concentrated risks, mental health and homelessness were not significantly associated with HIV/STI/HCV outcomes in bivariate associations or longitudinal analyses among all study participants.

This finding stands in direct contrast to prior peer-reviewed research that has identified significant bi- and multivariate cross-sectional and longitudinal relationships between homelessness, HIV/STI/HCV outcomes, and mental health risks among recently released populations [45,46]. The non-significant finding in the present study may be attributed to the timeframe examined in this study. As already noted, participants were recruited within one year of release and completed follow-up surveys 6-months post-baseline. Given the literature pointing to significant associations between “time since release” and worsening mental health conditions, housing instability, and HIV risk outcomes [47], it may be possible that the longer the window post-release, the more acute the mental health and homelessness risks faced, and the more deleterious the HIV/STI/HCV outcomes. Although additional research is needed in this regard, prior findings point to worsening outcomes the longer the post-release period among previously incarcerated populations with severe mental health risks [48]. In contrast, other studies have pointed to a lowering of health and behavioral health risks the longer women are out of jails and prisons—particularly with the provision of culturally appropriate, trauma-informed health interventions and support [47,49]. Similarly, results from an earlier study that utilized this current dataset suggest that longer periods of time since release are associated with significantly higher likelihood of engagement in Transition Clinic health services. This may indicate that these periods serve as a “critical time” for optimal engagement [36]. Taken together, these results highlight complex associations between the post-incarceration period and health, behavioral health, and homelessness. We recommend that additional studies with years-long follow-up windows be conducted to examine how threats to mental health, housing, and HIV/STI/HCV may change over time. Additional studies are also needed to examine how these risks may synergistically interact with other health and behavioral health risks, such as the SAVA syndemic (i.e., substance use and IPV).

Second, although this is the first study to examine typologies of HIV/STI/HCV risks among women recently released from incarceration using a SAVA syndemic or SAVA MH + H lens, our findings suggest that at least three SAVA MH + H typologies may exist. In comparison to Cluster 1, which exhibited the most concentrated SAVA MH + H risks, Cluster 2 reported virtually no drug use or alcohol misuse but had the highest rates of lifetime trauma and PTSD, as well as the second-highest percentage of unhoused women. Cluster 2 was also slightly larger than the first, had the smallest percentage of Black and Indigenous Women of Color (BIWOC), and had the highest educational attainment. In terms of SAVA MH + H syndemic risks, the third and final cluster (Cluster 3) reported far less hard drug use and alcohol misuse than their peers in Cluster 1, substantially less lifetime violence and sexual trauma, high-risk sexual encounters, higher rates of severe depression, the least PTSD, and the smallest homelessness risks. This latter group was also the largest, youngest, had the most BIWOC women, and was the least educated. Notably, while homelessness rates varied between groups and did not emerge as a specific HIV/STI/HCV risk in these data, all were fairly high, with a range of 32–47%, highlighting the need to address this quality of life and otherwise high-risk issue among this high-risk population.

These findings are revelatory and further underscore the continuing need to leverage the SAVA syndemic and other synergistic conceptualizations of risks in support of efforts to (1) develop HIV/STI/HCV interventions that target critical intragroup differences among women relative to the forms and types of transmission risks faced; and (2) end the co-occurring epidemics of HIV, other sexually transmitted infections, and HCV. Additional studies are needed to examine protective factors that may also exist. As we have noted, Cluster 2 had fewer women of color and more women with higher educational attainment. Women in this cluster engaged in little-to-no substance use/misuse, while navigating high rates of lifetime trauma, homelessness, and mental health risks. Additionally, while more women in Cluster 3 utilized drugs and alcohol, fewer among this group used/misused substances, and they were the least likely to be unhoused and engaged in few-to-no HIV/STI sexual risk behaviors. In view of this finding, we recommend that additional research be conducted to identify resilience factors in which women across clusters may be actively engaged in order to mitigate HIV/STI/HCV risks and prevent a downward slide into homelessness. The varied risks coinciding with racial variation highlight the importance of this factor in stratification and potential reduction, and they are worthy of further study. Studies suggest that criminal/legal-system-involved women engage in a highly intentional set of risk mitigation and navigation processes that, if better understood and supported, could be leveraged to prevent HIV/STI/HCV transmission and related risks, such as a transition into homelessness [50]. These strategies are noted to include “secondary abstinence” and condom usage negotiation, and they may be informed by faith-based beliefs and the use of community support [51].

Third, and finally, in light of the differences in the numbers of women engaging in/experiencing different forms of HIV/STI/HCV risk behaviors across the three clusters, it may also be true that without immediate and appropriately tailored intervention, SAVA MH + H risks may increase. As noted, hard drug use was the only significant predictor of the HIV composite outcome in the logistic regression analyses, with hard drug users showing over four times the odds of HIV/STI/HCV risk compared to non-users. Coerced sex during childhood also showed a trend-level association with the outcome, with those having experienced coerced sex having over twice the odds for HIV/HCV/STI risks. While additional research is needed with a larger dataset, this finding might suggest that in addition to operating synergistically, SAVA MH + H risks may operate hierarchically and along a continuum, with hard drug use representing a proximal risk during the weeks and months post-transition. While the prior literature points to a web of HIV/HCV/STI risks faced concurrently by women emerging from carceral settings [47], and these risks operate hierarchically [52], virtually nothing is known regarding how, why, and under what circumstances risks may increase in their level and severity of threat in the weeks, months, and years following re-entry. As an immediate next step, we recommend that additional longitudinal mixed-methods studies be conducted to examine HIV/HCV/STI risk pathways and risk progression through the lens of temporality, synergism, and hierarchy.

## 5. Limitations

The limitations of this study include the use of self-reported data. Still, data that are self-reported have repeatedly proven reliable in research studies [53]. Respondents across studies have shown a high recall for health-related outcomes as well as HIV [54].

This use of cross-sectional data for cluster modeling also means that significant associations cannot be determined. As a result, the identified SAVA MH + H syndemic risk clusters must be interpreted cautiously. As indicated above, we recommend that additional studies be conducted to advance our understanding of how the variables under consideration in this study may work in concert to predict SAVA MH + H syndemic risk behaviors and HIV/STI/HCV outcomes among women transitioning out of jails and prisons. In addition, the English-speaking requirement to participate may have limited the number of Latina participants, as some may have been less proficient in English. Due to sample size limitations, it was not possible to create interaction terms to examine synergistic interactions between the SAVA MH + H risk variables. Future studies with larger sample sizes are needed to examine possible interactions. Lastly, generalizations related to substance use to the larger population of formerly incarcerated women should be made cautiously, given that the recruitment criteria were based on self-reports of recent alcohol use or SUDs. As a largely exploratory study, no correction for multiple testing was applied; thus, the results should be regarded as tentative and in need of replication.

## 6. Conclusions

The findings from this study offer important insight regarding SAVA syndemic, mental health, and homelessness HIV/STI/HCV risk clusters, as well as the hierarchies of HIV/STI/HCV risks experienced by women recently released from incarceration during the highly vulnerable window post-transition. The results point to the need for tailored HIV/STI/HCV prevention and intervention services that recognize these important intragroup differences in the forms and amount of substance use, intimate partner violence, mental health, and homelessness risks experienced. To be optimally effective, HIV/STI/HCV prevention programs designed to prevent transmission among women in the U.S. legal system—a highly vulnerable population—must also prioritize substance use risk behaviors associated with the consumption of hard drugs in the weeks and months following release from incarceration.

## Figures and Tables

**Table 1 healthcare-11-01066-t001:** Cluster homogeneity.

Clusters	Pseudo-F
2	21.6
3	18.2
4	16.0
5	14.4
6	13.1

**Table 2 healthcare-11-01066-t002:** Expanded baseline SAVA syndemic risk clusters and HIV/STI/HCV outcomes.

Characteristic	Cluster 1 N = 34	Cluster 2 N = 45	Cluster 3 N = 74	Entire Sample N = 206 *
Sociodemographic characteristics				
Age 40 years or older	53%	44%	5%	35%
BIWOC race/ethnicity	53%	22%	59%	46%
Some college or greater education	32%	56%	19%	34%
Housing status				
Unhoused	47%	40%	32%	34%
Childhood trauma				
Physical abuse aged <16	68%	51%	41%	49%
Sex coerced aged <16	79%	67%	20%	50%
Sex forced aged <16	67%	40%	4%	30%
Other substance use				
Marijuana use past 90 days	59%	9%	36%	33%
Alcohol use past 90 days	76%	9%	41%	40%
Expanded SAVA syndemic risk clusters				
Substance Use				
Hard drug use past 90 days	97%	0%	28%	39%
Prescription drug use past 90 days	24%	4%	09%	13%
Intimate partner violence				
Physical abuse aged >16	85%	84%	57%	72%
Sex coerced aged >16	62%	71%	1%	33%
Sex forced aged >16	67%	73%	5%	38%
Mental health				
Severe depression	94%	58%	70%	75%
PTSD	91%	96%	74%	84%
HIV/STI/HCV outcome				
Unprotected sex past 30 days	88%	53%	30%	50%
Sex while high, with HIV+ partner, or with IDU partner, past 30 days	62%	13%	4%	20%

* N = 206 at baseline. Note: this is greater than the sum of Clusters 1, 2, and 3 (N = 153), due to the removal of boundary cases.

**Table 3 healthcare-11-01066-t003:** Longitudinal multivariate logistic regression expanded SAVA syndemic risks and HIV/STI/HCV outcomes.

HIV/STI/HCV Composite	Odds Ratio	Std. Err.	*p* > |z|	Lower 95% CI	Upper 95% CI
Sociodemographic characteristics					
Age 40 years or older	0.82	0.3617316	0.660	0.34	1.94
BIWOC race/ethnicity	0.79	0.3975951	0.651	0.30	2.11
Some college or greater education	1.30	0.59	0.561	0.53	3.17
Housing status					
Unhoused	1.90	0.8027072	0.126	0.83	4.35
Childhood trauma					
Physical abuse aged <16	0.95	0.4239064	0.925	0.40	2.28
Sex coerced aged <16	2.30	1.086514	0.077 ^+^	0.91	5.80
Sex forced aged <16	1.12	0.5329074	0.811	0.44	2.84
Other substance use					
Marijuana use (past 90 days)	0.69	0.3242343	0.430	0.27	1.73
Alcohol use (past 90 days)	1.65	0.760738	0.273	0.67	4.07
Expanded SAVA syndemic risk clusters					
Substance use					
Hard drug use (past 90 days)	4.32	2.087291	0.002 **	1.67	11.13
Prescription drug use (past 90 days)	1.16	0.7085623	0.796	0.35	3.83
Intimate partner violence					
Physical abuse aged >16	2.64	1.552898	0.097	0.83	8.35
Sex coerced aged >16	0.69	0.3506522	0.472	0.25	1.86
Sex forced aged >16	1.70	0.7946392	0.251	0.68	4.25
Mental health					
Severe depression	1.17	0.6326619	0.770	0.40	3.37
PTSD positive	2.37	1.767297	0.244	0.55	10.20

** *p* < 0.01; ^+^ *p* < 0.1.

## Data Availability

The raw data supporting the conclusions of this article can be made available by the authors upon request, without undue reservation.

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
