# Peer review of "HIV/STI/HCV Risk Clusters and Hierarchies Experienced by Women Recently Released from Incarceration"

_healthcare, 2023, doi:10.3390/healthcare11081066_

Round 1
Reviewer 1 Report
The manuscript of Karen A. Johnson and co-authors, submitted to Int. J. of Environment Research and Public Health, is entitled “HIV/STI/HCV Risk Clusters and Hierarchies Experienced by Women Recently Released from Incarceration”.
Women recently released from incarceration in the US have the highest prevalence of SUD and IPV, as well as alcohol use disorder. The rate of mental health disorders and homelessness are also high in this group. The study used an expanded SAVA syndemic conceptual framework that included mental health and homelessness to identify distinct clusters of common characteristics. Cross-sectional data were collected from women participating in WORTH Transitions program. Cluster analysis and logistic regression methods were used. Three clusters were identified, and hard drug use was found to be a predictor of HIV/STI/HCV risks. The conclusion is made that HIV/STI/HCV prevention and intervention services should differentially target identified risk clusters and hard drug use in the studied group.
Critique:
Lines 132 – 140 should be deleted.
2.2 Recruitment and Enrollment: the size of the studied group (206 at baseline and 101 at 6-month follow-up surveys) seems to be small for the self-reported risk behaviors. The retention rate is also low.
Table 2: N=178 at baseline; compare with Line 161: 206 participants at baseline.
Lines 326-327: N=178 is not less than N=153.
Lines 338 and 348: “Logistical regression” should be changed to “Logistic regression”.
In Table 3 the whole numbers in the last column and 0 p-value in the last line raise questions.
Line 369: the fact that the authors did not find the significant association does not exclude that such association may exist.
The purpose of statistical study is not completely clear. Statistical results should be presented and interpreted more accurately, i.e. Table 3 should be fixed and results should be interpreted using Bonferroni correction method.
Taking into account the experimental design, the Bonferroni correction should be used to interpret the results obtained by the logistic regression analysis correctly. The p-value 0.002 is still below the Bonferroni corrected bound 0.05/18=0.028, but it is very close to the bound, so the reliability of the conclusion is not so high as it follows from the text of the manuscript.
Author Response
Please see details in the attached file.

Reviewer 2 Report
The authors conducted a study of HIV/STI/HCV risk using the SAVA syndemic framework. Overall the study is well-written and informative. I have recommendations primarily to improve the clarity of the manuscript.
1. How were non-Black/White racial measures handled in the analysis? Were they dropped altogether, or omitted from specific analyses?
2. More clarity on how covariates were utilized (related to the clusters) would be helpful.
3. Statistical tests should be included in your table 2 (testing for statistical differences in factors between clusters).
4. Information on multicollinearity assessment (e.g., a variance inflation factor) will be helpful as well, particularly given how closely related some of the model terms are.
5. The last row of Table 3 has "_cons |", what was this meant to refer to?
6. The discussion is overall well written; I only recommend adding more interpretation of racial/ethnic disparities, including racism as a stressor that can lead to substance use (this is particularly important given the race-related cluster findings).
Author Response
Please see responses in the attached file.

Round 2
Reviewer 2 Report
The authors have addressed my concerns.